# Tierra Del Fuego: What Is Left from the Precolonial Male Lineages?

**DOI:** 10.3390/genes13101712

**Published:** 2022-09-23

**Authors:** Pedro Rodrigues, Irina Florencia Velázquez, Julyana Ribeiro, Filipa Simão, António Amorim, Elizeu F. Carvalho, Claudio Marcelo Bravi, Néstor Guillermo Basso, Luciano Esteban Real, Claudio Galli, Andrea del Carmen González, Ariana Gamulin, Romina Saldutti, Maria Laura Parolin, Verónica Gomes, Leonor Gusmão

**Affiliations:** 1Instituto de Investigação e Inovação em Saúde, Universidade do Porto, 4099-002 Porto, Portugal; 2Institute of Pathology and Molecular Immunology, University of Porto (IPATIMUP), 4099-002 Porto, Portugal; 3Instituto de Diversidad y Evolución Austral (IDEAus-CONICET), Puerto Madryn 9120, Argentina; 4DNA Diagnostic Laboratory, State University of Rio de Janeiro, Rio de Janeiro 20550-013, Brazil; 5Faculty of Sciences, University of Porto, 4169-007 Porto, Portugal; 6Laboratorio de Genética Molecular Poblacional, IMBICE (CCT-CONICET, CIC-PBA), Universidad Nacional de La Plata (UNLP), La Plata 1906, Argentina; 7Servicio de Hemoterapia, Hospital Regional de Rio Grande, Rio Grande 9420, Argentina; 8Servicio de Hemoterapia, CEMEP, Rio Grande 9420, Argentina; 9Servicio de Hemoterapia, Hospital Regional de Ushuaia, Ushuaia 9410, Argentina; 10Servicio de Hemoterapia, Clínica San Jorge, Ushuaia 9410, Argentina

**Keywords:** Y chromosome, Y-STRs, Y-SNPs, Argentina, South America, admixed population

## Abstract

Similar to other South American regions, Tierra del Fuego has an admixed population characterized by distinct ancestors: Native Americans who first occupied the continent, European settlers who arrived from the late 15th century onwards, and Sub-Saharan Africans who were brought to the Americas for slave labor. To disclose the paternal lineages in the current population from Tierra del Fuego, 196 unrelated males were genotyped for 23 Y-STRs and 52 Y-SNPs. Haplotype and haplogroup diversities were high, indicating the absence of strong founder or drift events. A high frequency of Eurasian haplogroups was detected (94.4%), followed by Native American (5.1%) and African (0.5%) ones. The haplogroup R was the most abundant (48.5%), with the sub-haplogroup R-S116* taking up a quarter of the total dataset. Comparative analyses with other Latin American populations showed similarities with other admixed populations from Argentina. Regarding Eurasian populations, Tierra del Fuego presented similarities with Italian and Iberian populations. In an in-depth analysis of the haplogroup R-M269 and its subtypes, Tierra del Fuego displayed a close proximity to the Iberian Peninsula. The results from this study are in line with the historical records and reflect the severe demographic change led mainly by male newcomers with paternal European origin.

## 1. Introduction

Tierra del Fuego is a Patagonian archipelago located in the southernmost part of the American continent. It is formed by a main island (Isla Grande of Tierra del Fuego) and a group of smaller islands shared by Argentina and Chile (Appendix A).

Archaeological evidence places the arrival of Native Americans in the archipelago between 12,770 and 12,400 YBP [1]. The crossing of the first settlers from mainland to Tierra del Fuego and the gradual dispersal process would have favored the differentiation of this former population into different ethnic groups known as Selk’nam, Yagán, Kawésqar, and Haush [2,3,4].

Upon the European arrival in the Strait of Magellan in 1520, numerous contacts were forged with the Fuegian natives, but the earliest establishment of European settlers in Tierra del Fuego only occurred in 1869 when an Anglican mission was stationed in the present-day city of Ushuaia. After the border treaty between Argentina and Chile was signed, in 1881, the Argentine state handed over enormous extensions of land, and several new settlers moved into Tierra del Fuego, engaging mainly in sheep farming and gold mining [5]. This colonizing period was characterized by extreme violence towards the indigenous populations, where clashes between the natives and the newcomers, manhunts, and exogenous diseases exterminated most of the natives [4].

Some of these newcomers were immigrants, mainly from Europe, who moved overseas during the late 19th and early 20th centuries. Throughout this period, Argentina was the South American country that welcomed the largest number of immigrants who inherently impacted the Argentine demography. The bulk of these immigrants who settled in the country were from Italy and Spain, and to a lesser extent from other regions such as France, Germany, Russia, Croatia, the British Isles, and the Ottoman Empire [6].

Until the 1970s, the main migratory flows to Argentine Tierra del Fuego stemmed from Chile, probably due to its geographical proximity and labor shortage in the archipelago. To reverse this situation, Argentina launched a financial promotion regime to encourage Argentine citizens to move to the extreme south. As a result, thousands of migrants, mainly from central Argentina, settled in the archipelago between 1980 and 2010, leading to an increase in Tierra del Fuego population from 27,463 to 127,205 [7,8]. In the last Argentine census, in 2010, the proportion of internal migrants represented 61.6%, the vast majority coming from the Patagonian and central regions. The proportion of immigrants moving to Tierra del Fuego represented only 8.6%, arriving primarily from Chile, and to a lesser extent from Bolivia and Paraguay. In the last decades, Asian and African immigration in Tierra del Fuego has been scarce and European immigration does not stand out as it formerly did [9].

Previous studies on the genetic composition of urban populations show a heterogeneity in the migratory flows in the different regions of Argentina. Regarding mtDNA, Native American lineages reach the highest frequencies in the northern provinces [10,11,12,13], whereas typical European lineages were mainly reported in the central provinces [14,15]. Studies on Y-chromosomal markers evidence the overwhelming prevalence of Eurasian lineages in Argentina, except for the northwestern provinces, where the proportion of Native American paternal lineages is relatively high (27.8–49%) [15,16,17,18].

Argentine Patagonia follows the same trend. Studies in central Patagonia reported a conspicuous predominance of Native American maternal lineages in different populations [19,20,21,22,23,24]. In contrast, analyses on Y-chromosomal STRs (Y-STRs) and SNPs (Y-SNPs) in central Patagonia show a prevalence of Eurasian paternal lineages, which is even higher in the coastal towns of Puerto Madryn, Trelew, and Comodoro Rivadavia than in the Andean populations of Esquel and San Carlos de Bariloche (Appendix A) [24,25].

To evaluate the impact of migratory flows and the admixture processes, we investigated the origin of the paternal lineages present in the current population from Tierra del Fuego, through the analysis of Y-STRs and Y-SNPs. The results were interpreted considering the genealogical, historical, and demographic information available, to understand the impacts of the influx from other regions of Argentina, South America, Europe, and the Middle East, to the archipelago that was once exclusively inhabited by Native Americans. Furthermore, due to the large-scale Eurasian immigration to Argentina, the R-M269 lineages were analyzed in finer detail to estimate contributions from different European and Middle Eastern regions to the Tierra del Fuego paternal genetic background.

## 2. Materials and Methods

### 2.1. Sampling

A total of 196 samples were collected from unrelated males living in the cities of Río Grande and Ushuaia, Tierra del Fuego province, Argentina (Appendix A). The samples were obtained by mouth swabs and venipuncture, from blood donors who spontaneously attended hemotherapy services of public and private hospitals in both cities. DNA was extracted using a standard phenol–chloroform method.

Participants were informed of the scopes of the study and filled out a questionnaire to obtain information about their birthplace and from the three preceding generations (parents, grandparents, and great-grandparents), accompanied by a written informed consent. A summary of the information collected is available in the Appendix A.

The project and sample collection were approved by the Ethics Committee of the Investigações Biomédicas (IMBICE) under resolution number RENISCE000023 Resol/120618 and by the Teaching and Research Committees of the Regional Hospital of Río Grande and the Regional Hospital of Ushuaia.

### 2.2. Y Chromosome STR Typing

The samples were genotyped for 23 Y-STR loci using the PowerPlex^®^ Y23 System (Promega Corporation, Madison, WI, USA). Amplification was performed according to the manufacturer’s recommended protocol, and PCR products were separated and detected on an ABI3500 Genetic Analyzer (Applied Biosystems, Carlsbad, CA, USA). Genotyping was performed using the GeneMapper™ Software v5.0 (Applied Biosystems), and alleles were assigned based on the allelic ladder provided with the kit.

### 2.3. Y Chromosome SNP Typing

A total of 52 Y-SNPs were selected to discriminate the most frequent Eurasian, Native American, and Sub-Saharan African haplogroups (Appendix A). These Y-SNPs were assigned hierarchically over seven PCR/SNaPshot multiplexes and one singleplex (for V88 marker). To select the most appropriate multiplex for each sample, haplogroups were predicted based on the Y-STR haplotype data using NEVGEN (https://www.nevgen.org/, accessed on 30 October 2020).

The multiplexes were carried out through PCR and single-base extension (SBE) reaction using the SNaPshot™ Multiplex kit (Applied Biosystems), as previously described by Brion et al. [26] (Mx 1 and Mx 2), Gomes et al. [27] (Mx E2), Campos [28] (Mx E1), Resque et al. [29] (Mx R1 and Mx R2), and Aragão [30] (Mx Q). Details about the multiplexes are available in Appendix A.

The SNaPshot products were submitted to capillary electrophoresis on an ABI 3500 Genetic Analyzer (Applied Biosystems) and the results were analyzed using the GeneMapper™ Software v5.0 (Applied Biosystems).

The V88 was genotyped via Sanger sequencing method using the BigDye™ Terminator v3.1 Cycle Sequencing Kit (Applied Biosystems), as previously described in González et al. [31]. The products were run on an ABI 3500 Genetic Analyzer (Applied Biosystems) and analyzed using the Sequencing Analysis Software v6.0 (Applied Biosystems).

### 2.4. Data Analyses

Haplogroup frequencies were calculated by direct counting. Genetic diversities and pairwise *R_ST_* and *F_ST_* genetic distances between samples from Tierra del Fuego and other Latin American, African, and Eurasian populations were performed using the Arlequin 3.5.2.2 software [32]. Pairwise genetic distances were represented in a multidimensional scaling (MDS) plot using the software STATISTICA 10.0.0.15 (TIBCO Software Inc., Palo Alto, CA, USA). This software was also used to perform principal component analysis (PCA), to determine the contribution of each haplogroup to the separation of the populations under analysis.

Networks were constructed using the Network 10.0 software (Fluxos Technology Ltd., Colchester, UK). The reduced median [33] and the median-joining [34] methods were applied sequentially to resolve extensive reticulation. Differential microsatellite weighting inversely proportional to their variance was used to obtain the most parsimonious network in accordance with Qamar et al. [35].

The Eurasian contribution in Tierra del Fuego was estimated for R-M269 and its sub-haplogroups using the ADMIX 2.0 software [36]. This software calculates the admixture coefficient expressed by Bertorelle and Excoffier [37] based on the molecular information of a given number of parental populations. The coefficients of admixture were calculated considering distances between haplogroups, estimated as the number of differences, and a mutation rate of 8.71 × 10^−10^ [38]. All the runs were performed fitting 1000 random bootstrap samples. The admixture coefficients obtained by the program were translated into percentage values.

## 3. Results and Discussion

### 3.1. Tierra Del Fuego Genetic Diversity

A total of 193 different haplotypes were identified in the 196 samples analyzed (Appendix A). Out of these haplotypes, 190 are singletons and 3 are shared by 2 individuals. Samples with shared haplotypes belong to the same haplogroup (two pairs are from haplogroup R-S116* and one pair from haplogroup J-M172). Regarding Y-SNP data, 24 different haplogroups were identified (Figure 1). For both Y-STR haplotype and Y-SNP haplogroup data, high levels of diversity were obtained when compared to other Latin American admixed populations [29,39,40,41,42,43,44,45,46] denoting the absence of important founder or drift events on the Tierra del Fuego population (Appendix A).

### 3.2. Tierra Del Fuego Haplogroups

Haplogroups inside the clade R are the most widely represented in the Tierra del Fuego, accounting for nearly half of the dataset. The remaining haplogroups belong to clades J, E, I, G, Q, and T, and to the paragroup KL.

The most likely continental provenance of each of these haplogroups was traced to Eurasia, Native America, and Africa, based on the distribution of the Y-chromosomal haplogroups across different populations and considering the history of Tierra del Fuego as well as Argentina.

While some haplogroups in our sample are common on a single continent, others have a wider distribution, being present in both Eurasia and Africa. According to the history of Tierra del Fuego, the Eurasian component is thought to have been contributed by Spanish colonizers and immigrants with European and Middle Eastern ancestry. The African component is likely the result of the transatlantic slave trade that brought a significant number of enslaved people from West and Central Africa and, to a lesser extent, from Mozambique, to the ports of Buenos Aires [47,48].

The haplogroups whose geographic distribution is in both Eurasia and Africa are as follows: E-M35*, E-M78, E-M81, E-M123, J-12f2a*, J-M172, R-V88, T-M70, and KL-M9*. However, some of these haplogroups on the African continent are constrained to North and East Africa. A gene influx coming directly from those regions is rather unlikely and, consequently, haplogroups E-M78, E-M81, E-M123, J-12f2a*, J-M172, T-M70, and KL-M9* were regarded as being brought to South America by Eurasians.

For haplogroups E-M35* and R-V88, which are present in African regions involved in the slave trade, networks were set up using haplotypes from African and Eurasian populations.

Although at low frequencies [49,50,51], the haplogroup E-M35* has been commonly reported in West and Central Africa [52,53,54]. Outside Africa, E-M35* is found at low frequencies in the European continent, being confined to Southern Europe [49,51,55,56]. In a network built with both African and European reference samples (Appendix A) [52,53,54,55,56,57,58], despite not sharing haplotype with any other sample, the E-M35* sample from Tierra del Fuego is closer to the Eurasian than to the African haplotypes, suggesting a Eurasian source for this chromosome. Genealogical data also support a Eurasian provenance since this individual’s great-grandfather was from Austria (see Appendix A).

Haplogroup R-V88, found in two individuals from Tierra del Fuego, is chiefly reported in Africa, more precisely in the Sahel region [59,60,61,62]. Despite being rare in Europe, R-V88 was once diffused across the southern regions of the continent [63] but was gradually replaced by other haplogroups coming from Asia [61]. Nowadays, haplogroup R-V88 is still often reported on Sardinia (Italy) and Corsica (France) islands, virtually the last places in Europe preserving this lineage [60,64,65,66]. In a network assembled using African and European haplotypes, one of the two R-V88 chromosomes presents a similar haplotype to European samples (Appendix A) [31,52,53,54,56,59,67,68]. The other R-V88 haplotype shows no clear similarities to the European and African haplotypes. Therefore, we could not pinpoint its continental provenance based on the network. Nevertheless, looking at the genealogical data retrieved from the donor, a European provenance seems more likely than an African one, since his great-grandfather was Italian (Appendix A). Thus, both R-V88 chromosomes were considered to have been carried to Argentina from Eurasia.

### 3.3. Tierra Del Fuego Continental Ancestry

The Eurasian lineages found in the Fuegian pool are included into the clades E (except for one sample belonging to haplogroup E-U290), G, I, J, K, T, and R (Figure 1). These lineages would have been brought in an early stage by Spanish colonizers during the colonial period, and, from the late 19th century on, by European, Middle Eastern, and Latin American immigrants.

Only two Native American haplogroups were reported in the present study, Q-M3* (xM19, Z19319, Z19483, M557, SA05) and Q-Z19483, both present at low frequency (3.06% and 2.04%, respectively). The Q-M3* is the founder of most lineages in the Americas that fall under the haplogroup Q. Its presence can be detected all over the double-continent [69,70,71,72]. Meanwhile, haplogroup Q-Z19483 is mostly distributed across the Central Andes, being likely associated with a male expansion through the mountain range that possibly took place during the Inca period [73,74].

It is worth noting that 8 of the 10 chromosomes carrying native haplogroups (those for which information about ancestors is available; see Appendix A) do not originate from Tierra del Fuego but instead came from Bolivia or other regions of Argentina.

The Q-M3* haplotypes found in Tierra del Fuego are similar to the ones found in Native Americans from northern Argentina and Patagonia (Appendix A) [75,76,77]. Meanwhile, the Q-Z19483 haplotype clusters with those from Bolivian and Peruvian natives. Therefore, it is likely that this haplogroup was brought to the archipelago chiefly by Bolivian and Peruvian immigrants.

The African component of the Tierra del Fuego sample only consists of one chromosome belonging to haplogroup E-U209. This typical Sub-Saharan African lineage is found in large proportions in West and Central Africa [53,78,79] and at lower frequencies in Southern Africa [78]. Outside the African continent, E-U290 is commonly found in Afro-American populations [80,81]. According to the genealogical information available (see Appendix A), the African Y chromosome found in our sample was brought to Tierra del Fuego by an immigrant from Paraguay.

According to haplogroups’ continental allocation, Tierra del Fuego presents a major Eurasian male contribution in its gene pool. Meanwhile, typical Native American and African haplogroups correspond to only a minor fraction (Figure 1). This continental ancestry pattern is identical to those previously described for Argentina, especially central and northeastern regions (Figure 2). Northwestern Argentina and northern and central Patagonia, on the other hand, show noticeably different values due to a higher incidence of Native American lineages [14,15,24].

Although Eurasian ancestry prevails in most admixed populations from Latin America [13,14,29,39,45,82,83,84,85,86,87,88,89,90,91,92,93], the studied sample of the current population of Tierra del Fuego is among those reported with the highest proportions (Figure 2).

The prevalence of Eurasian male lineages in Patagonia and other Latin America admixed populations is related to a sex-biased admixture. Throughout the Spanish colonial period, in addition to a predominance of Spanish men arriving to the Americas, mating between European males and Native American females was higher than between Native American males and European females [14,16,88,94].

Therefore, the analysis of mtDNA in the studied sample from Tierra del Fuego is expected to reveal a lower maternal than paternal Eurasian contribution. In fact, when we analyzed the genealogical data of our sample, we observed a lower proportion of matrilineal grandmothers coming from Eurasia than patrilineal grandfathers. While 76% of the individuals included in our sample report having a paternal great-grandfather who migrated from countries outside of America (Appendix A), this percentage drops to 46% in relation to maternal grandmothers. 

This phenomenon led to a reduced number of Native American male lineages in Latin American admixed populations, as observed in Tierra del Fuego. The unfriendly arrival of new settlers to the archipelago, which led to a drastic decline of the native population, may also have contributed to the small number of native paternal lineages present in the dataset. The low frequency of native paternal lineages present in the dataset also stems from a resettling, in the 1970s, caused by labor and economic incentives from the Argentinian government. This prompted an extensive migratory flow mainly from central Argentina, where the frequency of Eurasian paternal lineages is remarkably high (Figure 2), to the extreme south of the country [5,7,8].

As abovementioned, during the colonial period, Argentina received a considerable influx of enslaved Africans. The impact of this African influx was so high that according to the 1836 census, 26% of the residents of Buenos Aires were of African descent [95]. However, in studies on Y-chromosomal lineages from Argentine modern population, the African lineages are constantly underrepresented (Figure 2). The causes behind this decline have historically been attributed to yellow fever, high infant mortality, wars, and migrations to Uruguay [96]. Currently, it is thought that a biased mating between African woman and European-descent men was the main cause for this Afro-Argentine population drop [96].

### 3.4. Comparisons with Latin American Populations

Pairwise *F_ST_* genetic distances using Y-SNPs and pairwise *R_ST_* genetic distances based on 21 Y-STRs (all loci from PowerPlex^®^ Y23 kit except DYS385a/b) were calculated between our dataset and other admixed Latin American populations (see Appendix A), in order to evaluate similarities of those populations with Tierra del Fuego. Reference populations (Native Americans from Bolivia and Argentina, Yoruba people from Nigeria, Equatoguineans, and Spanish) were added to the analyses.

Tierra del Fuego only presents nonstatistically significant *R_ST_* values (*p* > 0.05) with Argentina (Appendix A). For pairwise *F_ST_* genetic distances, Tierra del Fuego also exhibits nonstatistically significant values with northern Brazil, central–west Brazil, southeastern Brazil, southern Brazil, and Nicaragua (Appendix A). In the MDS analysis, Tierra del Fuego plots closer to Argentina than central Patagonia. As discussed earlier, the continental origin of the lineages sampled in Tierra del Fuego presents higher similarities to the overall Argentinian population than other regions of Patagonia. In comparison with Tierra del Fuego, central Patagonia plots farther away from the reference Eurasian population (Spain) and closer to the reference Native American population (Bolivia_NativeAmerican), indicating a higher Native American input in the central Patagonian than in the Tierra del Fuego population.

For both MDS plots (Figure 3 and Figure 4) generated from the pairwise genetic distances calculations, most of the admixed populations, including Tierra del Fuego, tend to cluster together and plot near to the Spanish population, evidencing a higher proportion of Eurasian male lineages than Native American and African ones. Furthermore, the male lineages from the admixed Bolivian, Peruvian, and Mexican populations seem to be less affected by the Eurasian gene flow to the Americas, as already addressed in other studies [15,40,97,98], having shorter genetic distances to the native population compared to the other admixed populations under analysis.

The genealogical data collected evidence that the sample from Tierra del Fuego has a wide variety of geographic origins within Argentina. Of the 185 individuals born in Argentina, 110 reported central Argentine provinces as their birthplace, 49 in southern Argentina/Patagonia (of these, 43 were born in Tierra del Fuego), and 26 in northern Argentina (see Appendix A). Furthermore, based on paternal geographical origin, only 4 individuals have their origin in Patagonia, compared to 67 claiming their paternal origin in central Argentina and 20 in northern Argentina. This illustrates the reason why the modern population from Tierra del Fuego deviates from other Patagonian populations standing close to the overall Argentinian population.

Tierra del Fuego shares connections with the rest of Latin America, and some of the lineages in our dataset trace back to other Latin American countries. Out of the 196 sample donors, 21 indicated a Latin American origin, accounting for 10.7%, of which slightly more than half were from Chile (*n* = 11; 5.6%), followed by Bolivia (*n* = 4; 2.0%) and Paraguay (*n* = 3; 1.5%). Most of these lineages originating elsewhere in Latin America present a Eurasian origin. As addressed in the introduction, these three countries bordering Argentina were the main ones from where the immigrants in Tierra del Fuego had moved in the recent decades.

### 3.5. Comparisons with Eurasian Populations

As per the historical records from Tierra del Fuego and Argentina, the lineages within the Eurasian component of our dataset are likely to be predominantly from Europe and, to a lesser extent, from the Middle East. Hence, for the analyses that are discussed hereafter, several European and Middle Eastern populations were included. All the non-Eurasian lineages making up our dataset were not considered for the forthcoming analyses.

Pairwise *F_ST_* genetic distances using Y-SNPs and pairwise *R_ST_* genetic distances based on 21 Y-STRs (all loci from PowerPlex^®^ Y23 kit except DYS385a/b) were calculated between Tierra del Fuego and the Eurasian populations (Appendix A).

The comparative analysis using Y-SNPs revealed that Tierra del Fuego presents statistically nonsignificant (*p* > 0.05) values with the populations from northern Italy (*F_ST_* = 0.00099; *p* = 0.27621), central Italy (*F_ST_* = −0.00273; *p* = 0.78299), and Galicia (*F_ST_* = 0.01058; *p* = 0.05663), and the pairwise analysis using Y-STRs revealed nonsignificant values with northern Italy (*R_ST_* = 0.00089; *p* = 0.21493), Portugal (*R_ST_* = 0.00137; *p* = 0.21483), and Tyrol (*R_ST_* = 0.00365; *p* = 0.07514).

Analyzing these results, one can see that gene flow from Italy and the Iberian Peninsula strongly impacted the Fuegian male lineages. Indeed, the Italian and the Iberian (mainly Spanish) influxes in Argentina were notorious. It is estimated that roughly 2 million Italians and 1.4 million Spanish immigrants moved to Argentina between 1881 and 1914 [6,101], keeping in mind that prior to large-scale immigrations, Argentina had been subject to Spanish influxes throughout the nearly three centuries of Spanish colonization. Historical records point to a similar immigration from northern and southern Italy to Argentina [102]. However, in both MDS plots (Figure 5 and Figure 6), northern Italy seems to have played a more compelling role. The fact that Galicia is the Spanish region presenting the shortest pairwise *F_ST_* genetic distance with Tierra del Fuego (Figure 6) is not unexpected, since Galicia was the region from where the majority of Spanish immigrants departed [6,103,104,105].

Genealogical data collected from sample donors support a major Italian and Spanish influence in Tierra del Fuego. The 71 individuals (36.2%) who have a known European paternal heritage trace their origins mainly to Italy (*n* = 29; 14.8%) and Spain (*n* = 25; 12.8%) (Appendix A). The rest of Europe accounts for 8.7%, which includes France, Germany, Poland, Portugal, Ireland, Austria, the Czech Republic, and Russia. Additionally, six individuals (3.1%) declared a Middle Eastern paternal ancestry. For those individuals who provided paternal origins up to great-grandfathers, a predominance of European and Middle Eastern origins (76.1%) of the great-grandfathers is noticeable (Appendix A). As we analyze the provenances of the great-grandfathers, grandfathers, and fathers of the study subjects, we observe that the European and Middle Eastern provenances decrease from generation to generation. Most of the grandfathers and fathers were born in Argentina, contrasting markedly with the provenance of the great-grandfathers. The high European and Middle Eastern origin in great-grandfathers is most likely linked to the large-scale immigration period. After that migratory phenomenon, the European and Middle Eastern influx started to become less common, and, for that reason, one can observe the sharp difference across the different generations.

The haplogroup R is the most frequent in Tierra del Fuego, accounting for around half of the Eurasian lineages. The overriding bulk of these R lineages are subtypes of the haplogroup R-M269. These subtypes are broadly spread in both Europe and the Middle East, presenting a clinal distribution with increasing frequencies in Western Europe [112]. Unlike most of the R-M269 lineages, the upstream haplogroups, R-M269*(xL23) and R-L23*(xU106, S116), are uncommon in Western Europe, showing frequency peaks in Eastern Europe, Anatolia, Caucasus, and circum-Uralic communities [112,113]. The remaining downstream haplogroups are spread across Western Europe, displaying distinct geographic distributions. The haplogroup R-U106 is mostly reported in The Netherlands, Germany, Switzerland, Austria, Western Poland, and throughout Northern Europe [112,113,114,115,116]. The R-S116*(xU152, M529, M153, M167), the most prevalent lineage in Tierra del Fuego, is characteristically found all over Western Europe, peaking in Portugal and Spain [112]. Regarding the R-S116* sub-lineages, R-U152 presents the highest frequencies in Northern and Central Italy, France, and Switzerland [56,112,113,114,115,116,117], R-M529 in the British Isles and Brittany [112,113,115,117], R-M153 in the Basque Country, and R-M167 in Catalonia [55,118].

A PC analysis was performed to inquire which sub-haplogroups have the most weight in the differentiation of Tierra del Fuego, European, and Middle Eastern populations (Figure 7). In the lower right quadrant, the Celtic- and Germanic-speaking populations are observed. The Celtic-speaking populations are dominated by the haplogroup R-M529, while the Germanic ones, although some feature the haplogroup R-M529 in moderate frequencies, exhibit higher frequencies on haplogroup R-U106. Slavic-speaking countries and Turkey display high frequencies of haplogroups R-M269* (except Russia) and R-L23*. Moreover, Poland and Russia show significant frequencies of the subtype R-U106. In the upper left quadrant are the four Italian populations, all of which exhibit high frequencies for the subtype R-U152 and moderate ones for the subtype R-M269*. In the first quadrant are the population under study and the populations from Portugal, Spain, and France. All these populations present high frequencies for the sub-haplogroup R-S116*.

In order to estimate the contribution of Eurasian populations in Tierra del Fuego and other South American regions regarding R-M269 and its sub-haplogroups, admixture coefficients were computed using the ADMIX 2.0 software. For the analysis, populations from the Iberian Peninsula (Portugal and Spain) [112], Italy [112], Germany [112], Poland [112], England [112], Ireland [112], and Turkey [112] were considered as parental populations.

The results illustrated in Figure 8 show a major Iberian contribution in Tierra del Fuego, followed by Italy and Ireland. This is also supported by the PC analysis showing that the typical Iberian lineage, R-S116*, has the most weight in the Tierra del Fuego population. The German, English, and Turkish contributions are lower, and the Polish one is null. It is important to highlight that the frequencies of the haplogroup R-M269 and its subtypes observed in Ireland are similar to that in Scotland, Wales, and Brittany [112,113]; therefore, a provenance from these regions should not be ruled out.

Compared with the other South American populations under analysis, Tierra del Fuego stands out at a continental level, presenting the lowest Iberian and the highest Italian and Irish contributions. Since it was not possible to include other samples from Argentina in the analysis (due to lack of data), the pattern found may be present in other regions of the country. Because Tierra del Fuego reflects the overall Argentinian genetic background, the results obtained could also imply lower Spanish and higher Italian and Irish contributions in Argentina compared to the rest of South America, regarding R-M269 and its sub-lineages.

## 4. Conclusions

The paternal lineages of the current population from Tierra del Fuego present high haplotype and haplogroup diversities, pointing to the absence of strong founder or drift events. This paternal genetic legacy reflects the influence of the European gene flow into the Americas during the colonial period and large-scale immigrations, tallying one of the highest percentages of Eurasian lineages in Latin America. Furthermore, according to genealogical data, the fathers and grandfathers of the sample donors were mainly from central Argentina, where the Eurasian lineages score high frequencies. Therefore, Native American haplogroups are underrepresented in our sample, contrasting with central and northern Patagonia and northwestern Argentina, which feature higher frequencies of native lineages. Moreover, most of the Native American lineages found in Tierra del Fuego should be a result of influxes from other parts of Argentina, Bolivia, and Peru, denoting a lack of native Fuegian lineages in the modern population.

It is worth noting that this study aimed to assess the impact of migratory flows and admixture processes on the current population, and therefore samples were randomly selected, and genealogical information was used only to interpret the results, and not as a selection criterion. However, communities that descend from Fuegian ancestral populations still persist in Tierra del Fuego, in which a greater preservation of the native genetic pool is expected.

Typical Sub-Saharan African haplogroups score a scant percentage in Tierra del Fuego, a very similar result to that found in other Argentinian regions. This bolsters that the African influx left little trace on the paternal component of the Argentinian population, unlike what occurred in other Latin American regions, such as northeastern Brazil, Colombia, and Ecuador.

Pairwise genetic distances using haplotype data show that Tierra del Fuego has proximity to populations from northern Italy and the Iberian Peninsula, with the gap for the Italian population being perceptibly smaller. This indicates that Italian and Spanish inflows during the colonial period and large-scale immigrations markedly shaped the Y-chromosomal landscape of Tierra del Fuego. Analyses focused on the haplogroup R-M269 and its subtypes suggest a major Iberian contribution, followed by smaller inputs from Italy and the British Isles. Furthermore, the PC analysis shows a high influence of the sub-haplogroup R-S116* in Tierra del Fuego and Spain, meanwhile Italy and the British Isles are mainly influenced by R-U152 and R-M529, respectively. This is in line with the genealogical data, which shows that the individuals who trace back their paternal ancestry to Spain carry, mainly, the sub-haplogroup R-S116*, and the individuals with Italian ancestry bear mainly the sub-haplogroup R-U152.

## Figures and Tables

**Figure 1 genes-13-01712-f001:**
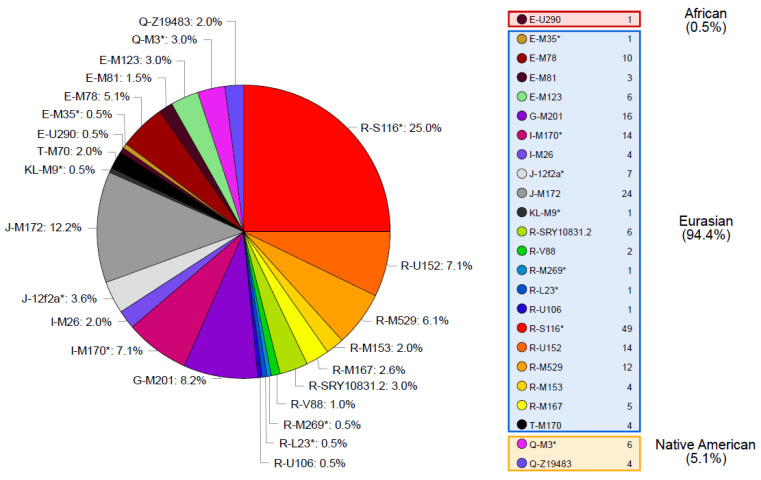
Y-chromosomal haplogroups identified in Tierra del Fuego and their absolute and relative frequencies in our sample.

**Figure 2 genes-13-01712-f002:**
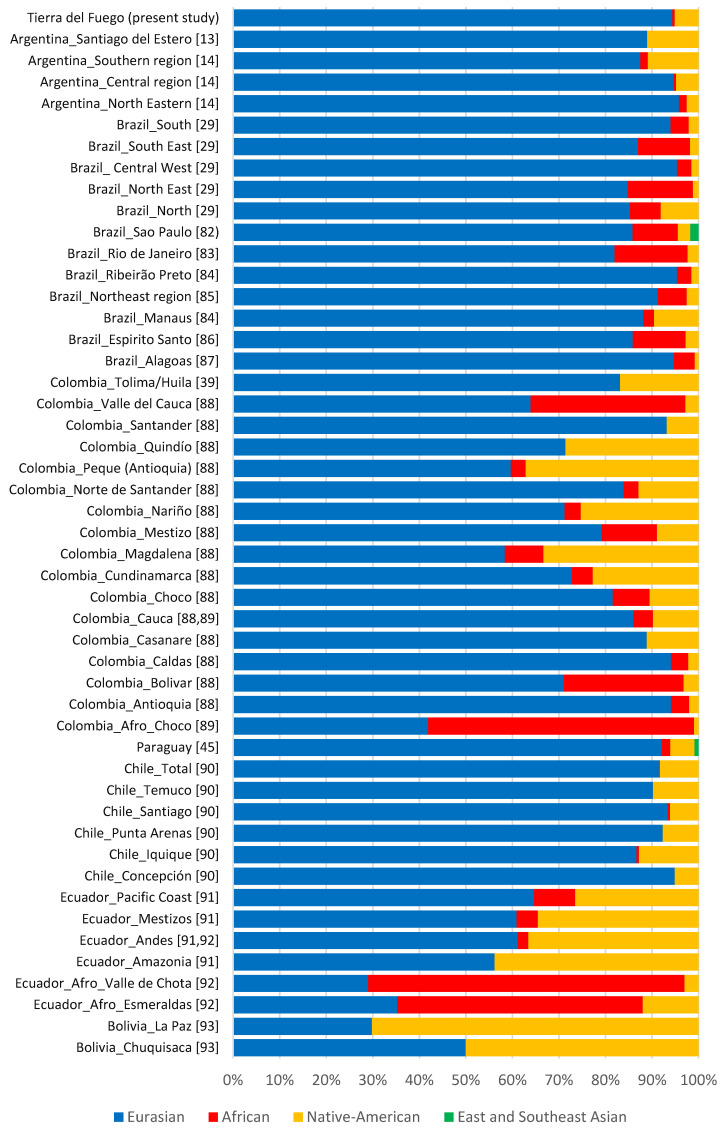
Paternal ancestry estimates on different South American populations [13,14,29,39,45,82,83,84,85,86,87,88,89,90,91,92,93].

**Figure 3 genes-13-01712-f003:**
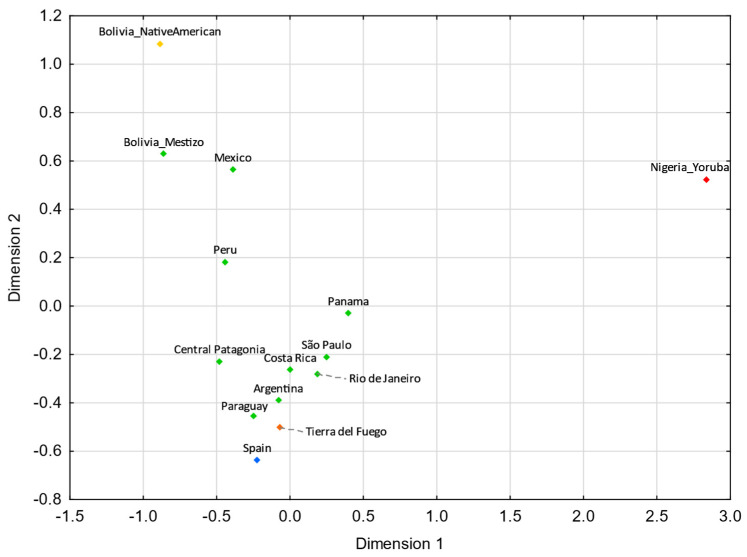
MDS plot based on *R_ST_* genetic distances between Tierra del Fuego and other Latin American populations [42,43,45,97], based on 21 Y-STRs (stress = 0.0109). Tierra del Fuego is plotted in orange and the other admixed populations of Latin America are depicted in green. The European [43,99], Native American [43], and African [43] populations are shown in blue, yellow, and red, respectively.

**Figure 4 genes-13-01712-f004:**
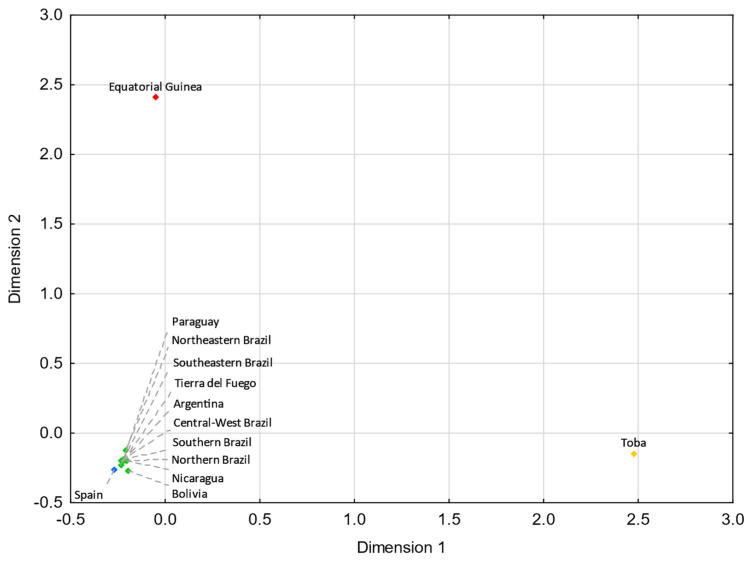
MDS plot based on *F_ST_* genetic distances between Tierra del Fuego and other Latin American populations [14,29,40,41,45], based on 14 haplogroups (A-M13, A4-SRY10831.1*(xM168), C-M130, E-M96*(xM35), E-M35, F-M213*(xM201, M170, 12f2a, M9), G-M201, I-M170, J-12f2a, KLNT-M9, Q-M242*(xM3), Q-M3, R-M207*(xM343), and R-M343) (stress = 0.0019). Tierra del Fuego is plotted in orange (not visible due to the clustering). The admixed populations of Latin America are depicted in green. The European [55], Native American [100], and African [31] populations are shown in blue, yellow, and red, respectively.

**Figure 5 genes-13-01712-f005:**
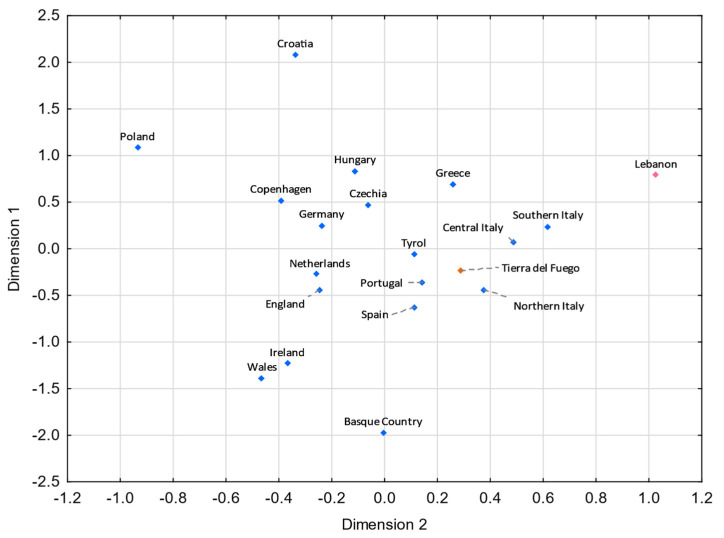
MDS plot based on *R_ST_* genetic distances between Tierra del Fuego and European and Middle Eastern populations [43,99,106], based on 21 Y-STRs common to all studies and considering only European and Middle Eastern lineages from Tierra del Fuego (stress = 0.0596). Tierra del Fuego is plotted in orange, and the European and Middle Eastern populations are shown in blue and pink, respectively.

**Figure 6 genes-13-01712-f006:**
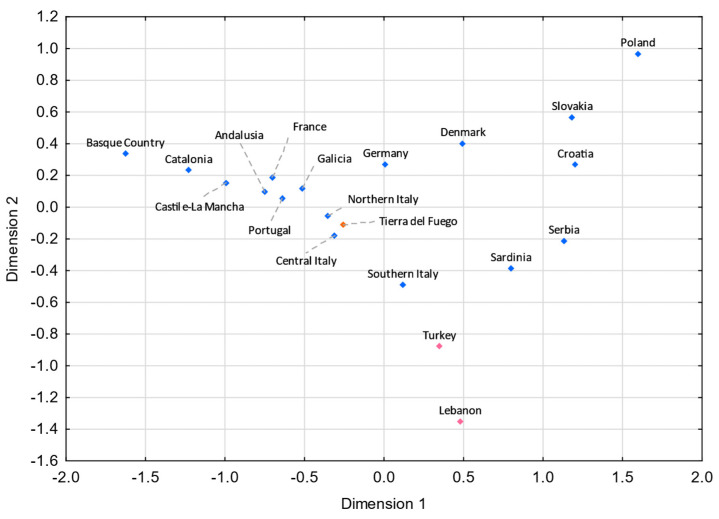
MDS plot based on the *F_ST_* genetic distances between Tierra del Fuego and European and Middle Eastern populations [55,56,57,58,107,108,109,110,111], based on 11 Eurasian haplogroups (E-M35*(xM78, M81, M123), E-M78, E-M81, E-M123, G-M201, I-M170, J-12f2a, KLNT-M9, R-M207*(xSRY10831.2, M343), R-SRY10831.2, and R-M343) (stress = 0.0671). Tierra del Fuego is plotted in orange. The European and Middle Eastern populations are shown in blue and pink, respectively.

**Figure 7 genes-13-01712-f007:**
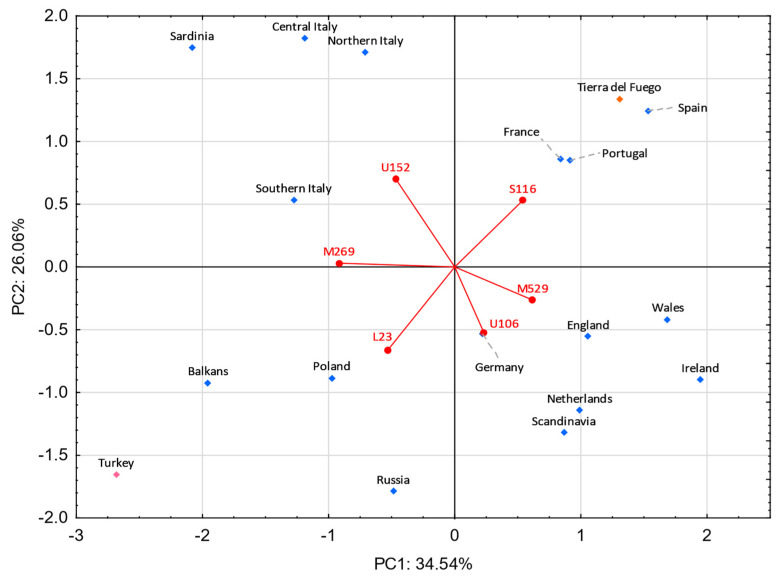
Principal component analysis based on R-M269 sub-haplogroups from Europe and the Middle East. Tierra del Fuego is plotted in orange. The European and Middle Eastern populations are shown in blue and pink, respectively. Populations used: Tierra del Fuego (present study), Balkans [112], England [112,113], France [112], Germany [112,113], Ireland [112,113], Poland [112,113], Portugal [112,113], Russia [112], Scandinavia [112], Spain [112], The Netherlands [112,119], Turkey [112,113], and four regions from Italy (northern Italy, central Italy, southern Italy, and Sardinia) [56].

**Figure 8 genes-13-01712-f008:**
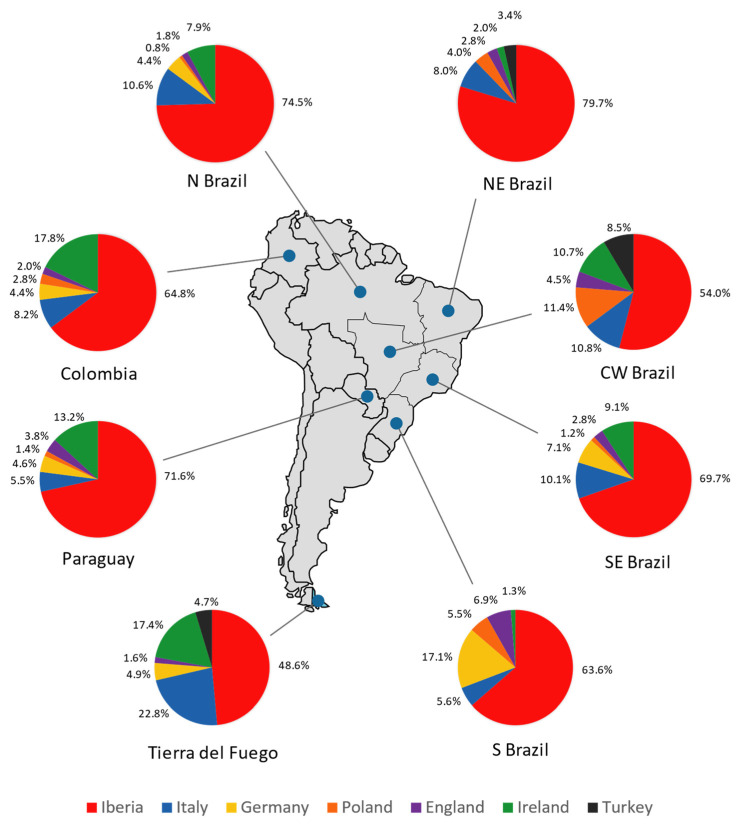
Pie charts representing the male contribution (in percentage) from the Iberian Peninsula, Italy, Germany, Poland, England, Ireland, and Turkey in Tierra del Fuego, Colombia, Paraguay, and the five geopolitical regions of Brazil (north, northeast, central–west, southeast, and south), obtained through ADMIX 2.0 analysis.

## Data Availability

The data presented in this study are available in the Appendix A.

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
