# Peer review of "Tierra Del Fuego: What Is Left from the Precolonial Male Lineages?"

_genes, 2022, doi:10.3390/genes13101712_

Round 1

Reviewer 1 Report

The study describes the genetic diversity of the population of Tierra del Fuego. It is based on standard analyzes of the Y chromosome (combination of STRs and SNPs), not new insights through genomic analyses. Therefore, overall merit of the publication is rather average, but still interesting as based on new dataset. 

Major comment:

Figure 1 is not available to review!

In the studies like this one, all results depend on the places, where the samples were secured. In fact, if the authors would look for the indigenous peoples in Tierra del Fuego, the percentage of the original Fuegian ancestry would be much higher. But if the samples were collected among the descendants of the immigrants, the result as obtained in their study is evident. The authors should describe this issue more carefully – for example how the places for the collection of the samples were chosen and whether the birthplaces recorded during sampling was important for the inclusion of not-inclusion of the individual in the study. Are there (in Tierra del Fuego) some other places where aboriginal (Amerindian) ancestry can be found in higher frequency than in Rio Grande and Ushuaia?

Minor comments:

Page 2 … “… thousands of migrants, mainly from central Argentina, settled in the archipelago, enhancing the Fuegian population growth rate”. Strictly (biologically) speaking it is not growth rate of the population, but immigration.

Figure 2 – do not use abbreviations for the ancestry colors. NAM might be read as Native American or North American … EUR is Eurasia or Europe? If Eurasia, why there ASIA is differentiated? In fact, there is enough place in the figure to write full text.

Sex-biased admixture is very interesting phenomenon in anthropology – is it possible to study it also in the population samples the authors dispose? Maybe something for maternal origin could be inferred from the info obtained during collections (birthplaces, provenance … etc.). If yes, such info might be included in Discussion.

Reviewer 2 Report

Rodrigues P et al carried out Y-STR typing and the determination using a 52 Y-SNP panel on 196 male samples from Tierra del Fuego. The detail about the migration of tested population was given and the results were well presented.  The investigation in this study will help clarify the population structure from paternal lineages. Because of a small sample size, the conclusion should be cautious.

1, Line 175-176,   2 individuals from each shared haplotypes are from the identical haplogroups?

2, The figure 1 is missing 

3, Line 266-268, "the highest values" might be misleading because of sampling and a small sample size.
